# OpenReview forum: "Can Large Language Models Develop Gambling Addiction?"
_ICLR.cc/2026/Conference — Submitted to ICLR 2026_

### Official Review · Reviewer_W9Tv · 2025-10-26

**Soundness:** 2
**Presentation:** 2
**Contribution:** 2
**Rating:** 2
**Confidence:** 4

**Summary:**

This paper investigates the gambling behavior of LLMs using a simulated slot machine with a negative expected value (30% win rate, 3x payout). A rational agent maximizing expected value should avoid this game, as prolonged play leads to almost certain loss. The authors find that 4 frontier LLMs (GPT-4o-mini, GPT-4.1-mini, Gemini-2.5-Flash, Claude-3.5-Haiku) deviate from this rational behavior, not only playing multiple times but occasionally going bust.

To predict this behavior, the authors construct a composite irrationality index (Eq. 1), a weighted linear combination of three heuristic behavioral metrics. This index shows a strong positive correlation with the observed bankruptcy rates (r-values between 0.770 and 0.933). Furthermore, they demonstrate that gambling behavior is moderated by the instruction prompt. Through an ablation study of five prompt components (goal-setting, reward maximization, hidden patterns, win-rate reminders, and win-reward reminders), they find that prompts containing more of these elements lead to a higher likelihood of the model going bust.

In the second part of the study, the authors analyze an open-weight model (Llama-3.1-8B). Using sparse autoencoders (SAEs), they identify "safe" and "risk" features within the Llama model's neural activations. They show that activating the "safe" features can steer the model's behavior, causing it to stop playing earlier and significantly reduce its bankruptcy rate.

**Strengths:**

The paper is accessible and tackles a highly relevant problem by studying gambling behavior in LLMs. The core behavioral results are interesting (i.e., systematic deviations from rationality in LLMs), and the authors successfully develop an index that effectively predicts the models' propensity to go bust, which has significant implications for behavioral analysis. They then strengthen these findings with a mechanistic investigation using SAEs, where causal interventions solidly connect the identified neural representations to the behavior.

**Weaknesses:**

My primary concern is that the work's contribution to understanding LLM behavior is somewhat limited in scope. The behavioral investigation is confined to a single, specific gambling scenario: a slot machine with a fixed 30% win rate and 3x payout. While this provides a valuable baseline, the scientific insights into LLMs' pathological gambling behavior at large remain constrained. To strengthen the generality of their findings, the authors should consider testing their hypotheses on at least one other gambling task (e.g., a different slot machine configuration or gambling devices other than slot machines) to evaluate how well the observed behaviors generalize across similar scenarios.

The neural intervention using SAEs to identify "safe" features is an interesting contribution. I acknowledge the practical necessity of using an open-weight model like Llama for this analysis. However, the steering results present a confusing contradiction with the prompting results from Experiment 1. Specifically, activating "safe" neural features in Llama reduces gambling, while providing closed-source models with more relevant information and goals in the prompt (Fig. 4) increases gambling. While this discrepancy could be attributed to the different model families used in each experiment, it points to a potentially unresolved issue. The paper would be strengthened by reconciling this mismatch, ideally by first replicating the key behavioral results (including the prompt complexity effect) on the Llama model to establish a consistent baseline.

**Questions:**

Why is the irrationality index constructed using the specific weights and the three heuristics indices in Eq. 1? This seems quite arbitrary and post-hoc if without strong theoretical justifications.

Why the behavioral evidence (Experiment 1) is primarily provided using close-source models while the neural evidence (Experiment 2) is primarily provided using the open-source Llama model? Why not conduct same behavioral experiments to Llama?

---

> ### Author Response · Authors · 2025-12-02
>
> ## Major Concern 1: Limited scope (single gambling scenario)
>
> We added an **Investment Choice experiment** to demonstrate cross-paradigm consistency. This experiment has a different structure from the slot machine, offering four options:
>
> - Option 1: safe exit with capital return
> - Option 2: 50% win probability, 1.8× payout
> - Option 3: 25% probability, 3.6× payout
> - Option 4: 10% probability, 9.0× payout
>
> Notably, Options 2, 3, and 4 have identical expected losses (-10%), allowing us to isolate risk preference from expected value computation.
>
> **Key Results:**
> - Goal-setting prompts increase bankruptcy from 40.0% to 74.6% (consistent with slot machine)
> - Variable betting produces +3.3% higher bankruptcy across all bet constraint levels (consistent with slot machine)
> - Option 4 selection increased from 15% (BASE) to 41% (GM)
>
> This cross-paradigm evidence demonstrates that our findings emerge from decision-making mechanisms rather than task-specific features.
>
> ---
>
> ## Major Concern 2: SAE vs Prompting contradiction
>
> Thank you for this excellent point. We have conducted and added behavioral experiments for LLaMA.
>
> Table 4 now includes LLaMA-3.1-8B results:
>
> - Fixed: 2.62% bankruptcy
> - Variable: 6.75% bankruptcy
>
> LLaMA also shows higher bankruptcy under variable betting than fixed betting, which is consistent with the pattern observed in closed-source models. Therefore, the SAE analysis finding that safe features increase stopping and risky features increase gambling **aligns with the behavioral patterns in the same model**.
>
> The apparent contradiction (activating safe features → reduces gambling; providing more information → increases gambling) can be interpreted as follows: The information provided through prompts (goal-setting, maximizing) creates conditions that activate risky features, which consequently increases gambling behavior. In other words, prompts are the **conditions** and neural features are the **mechanisms**.
>
> More specifically, our findings suggest the following causal pathway:
>
> 1. Goal-setting prompt provided
> 2. Increased activation of risky features in later layers (L24-L28)
> 3. Decreased stopping rate, increased gambling continuation
> 4. Increased bankruptcy rate
>
> This interpretation is also consistent with the word-feature analysis results showing that risky features associate with goal-pursuit language (goal, target, make, set).
>
> ---
>
> ## Minor Concern 1: Irrationality index weights arbitrary
>
> Eq.1 was originally designed to track how the tendency for irrational behavior changes depending on experimental conditions. However, in the process of creating a single integrated index, two limitations emerged: 1) the heuristic weights were not validated, and 2) the integration actually made it more difficult to understand which specific behavioral tendencies were exhibited.
>
> Therefore, in the revised manuscript, we **removed the composite index** and modified the text to examine each behavioral metric (I_BA, I_LC, I_EB) individually across experimental conditions:
>
> - I_BA: 0.14 (fixed) → 0.31 (variable), 2.3× increase
> - I_LC: 0.16 → 0.42, 2.7× increase
> - I_EB: 0.04 → 0.23, 6.4× increase
>
> We hope this modification better serves the original objective of tracking changes in irrationality.
>
> ---
>
> ## Minor Concern 2: Why different models for behavior vs neural
>
> We have conducted and added behavioral experiments for LLaMA. Thank you for this excellent suggestion.
>
> We now provide both behavioral and neural evidence for the same model (LLaMA-3.1-8B):
>
> - **Behavioral evidence**: LLaMA results included in Table 4 (Fixed: 2.62%, Variable: 6.75%)
> - **Neural evidence**: 6,400 games, 112 causal features identified (23 safe + 89 risky)

---

### Official Review · Reviewer_YnzE · 2025-10-30

**Soundness:** 2
**Presentation:** 2
**Contribution:** 2
**Rating:** 2
**Confidence:** 3

**Summary:**

The study explores LLMs' gambling behavior in a slot machine task, in which the authors identified addictive behavior with corresponding cognitive biases of LLMs, grounded in cognitive-behavioral patterns presented in human gambling addictions. To illuminate how LLMs internally represent risk in gambling, the author used Sparse Auto-encoder to discover features relevant to either safety or risk. Finally, through activation patching, the author steered model's behavior in a causal manner, validating the features extracted. Overall, the study shows LLMs' gambling addiction at different levels, pointing out a potential way of risk control in using LLMs as decision-makers.

**Strengths:**

1. The study defines LLMs' irrational behavior with rules that root in human addictive patterns, proposing a reliable way to quantify addiction in a betting task.
2. Fine-grained manipulations relevant to risk.
3. The layer-wise distributions of risk-related features are interesting.

**Weaknesses:**

1. Lack of norm. Humans mental disorders are diagnosed according to a norm basically, but for LLMs it seems tricky to have a norm.
2. Likewise, risky behavior (as indicated by the irrational index) in a gambling task might be insufficient in defining a gambling addiction as claimed in the paper.
3. Inconsistent model use: better investigate the same model on both behavior and neural level.
4. More controls are needed to make sure the neural features are only related to risk decision. When using activation patching, the specificity and generalizability of patching features should be checked in order to see if such patching undermines model performance or even induce more risk in other domains.

**Questions:**

1. Incorrect color labelling in Figure 1 (Phase 1, prompt composition), where G/M should be in red rather than green, according to the results in 3.2.
2. Does bankruptcy mean that the LLM run out of initial capital without quitting the game? Does the 'rounds effect' mean the average number of rounds LLMs play in a single game before quitting, or before bankruptcy? Please provide clear definition for all indicators used in paper.
3. In section 4, 6400 games mean Llama played 100 times in each condition? When extracting embeddings for SAE, which position within answers were used?
4. When identifying differential features between the safety and bankruptcy, how do you guarantee the specificity of the features? As the features can represent concept relevant to stop/leave/numeric values/... other than the safe-risky variable.
5. In finding 2 of activation patching, what does the safe context and risky context mean? What temperature did you use in the generation? As there are only 211 bankruptcy cases, -14.2% change means that there are still over 180 bankruptcy cases. Worth a check whether the change reflects noise.
6. Did Llama3.1-8b show similar patterns as found in phase 1? Such as different effect sizes caused by different prompt components. Did the 211 bankruptcy cases primarily come from a specific condition? Maybe changing a model will lead to a more balanced number of cases based on the variability shown in table 2.

---

> ### Author Response · Authors · 2025-12-02
>
> ## Major Concern 1: Lack of norm
>
> We thank the reviewer for raising this important methodological question. It allows us to clarify the objectives and approach of our research.
>
> Our study does not aim to diagnose individual LLMs as "addicted" or "not addicted." Instead, we seek to identify the **conditions under which addiction-like behavioral patterns emerge**. This is a study on condition effects and the identification of risk factors, not clinical diagnosis. Therefore, the absence of an absolute norm does not fundamentally constrain our research objectives.
>
> We addressed this issue systematically through a **within-model comparison** design:
>
> - The behavior of each model in the BASE condition serves as its own baseline
> - We do not ask "Is this model addicted?" but rather "**Does this condition significantly increase risky behavior compared to the baseline?**"
>
> This approach is robust in three aspects:
>
> 1. **Control**: Each model acts as its own control, accounting for baseline differences between models
> 2. **Measurement**: We measure behavioral changes induced by conditions rather than absolute levels
> 3. **Consistency**: Patterns consistently reproduced across six architecturally distinct models
>
> Interestingly, even human psychiatric diagnosis relies more on relative evaluation than absolute norms. The DSM-5 defines disorders based on "clinically significant distress or impairment" often relative to the individual's functional baseline (APA, 2013). Our within-model design is conceptually aligned with this principle.
>
> Our findings identify **actionable and preventable risk factors**:
>
> - Goal-setting prompts increase bankruptcy rate from 40.0% to 74.6%
> - Variable betting produces +3.3% higher bankruptcy across all bet constraint levels
>
> ---
>
> ## Major Concern 2: Insufficient definition (risky behavior vs gambling addiction)
>
>
> We appreciate the concern that risky behavior may be insufficient to define gambling addiction. To address this concern, we made two modifications.
>
> **First**, we clarified how irrationality is defined in the context of pathological gambling in behavioral psychology. Based on clinical literature (Petry 2005, Grant & Potenza 2006), we restructured our analysis around three core dimensions:
>
> 1. **Betting Aggressiveness**: Large or escalating bets relative to available resources
> 2. **Self-Regulation Failure**: Inability to adhere to self-imposed limits and rules
> 3. **Cognitive Distortions**: Unrealistic expectations including unattainable goal-setting or probability misjudgment
>
> **Second**, we added specific analyses measuring each dimension:
>
> - **Betting Aggressiveness**: I_BA, I_EB, Option 4 selection rate
> - **Self-Regulation Failure**: Goal escalation rate (56-59% in G/GM conditions)
> - **Cognitive Distortions**: Qualitative analysis through case studies (illusion of control, gambler's fallacy, loss chasing, house money effect)
>
> We removed the composite index (Eq.1) and separated individual metrics to allow clearer interpretation of the behavioral aspects captured by each.
>
> ---
>
> ## Major Concern 3: Inconsistent model use
>
>
> We have added behavioral experiment results for LLaMA. Table 4 now includes results for LLaMA-3.1-8B and Gemma-2-9B:
>
> | Model | Bet Type | Bankruptcy (%) |
> | :--- | :--- | :--- |
> | LLaMA-3.1-8B | Fixed | 2.62 ± 0.31 |
> | LLaMA-3.1-8B | Variable | **6.75** ± 0.56 |
> | Gemma-2-9B | Fixed | 12.81 ± 0.84 |
> | Gemma-2-9B | Variable | **29.06** ± 1.14 |
>
>
>
>
> Although the LLaMA model shows lower bankruptcy rates compared to closed proprietary models, **our core finding that autonomy influences irrational decision-making remains valid** (difference between fixed vs variable conditions).

---

> ### Author Response · Authors · 2025-12-02
>
> ## Minor Concerns
>
> **Figure 1 color labeling**
>
> We thank the reviewer for this keen observation. You are absolutely correct. Considering our findings that the G/M (Goal-setting/Maximizing) prompts reinforce risky decision-making, they should indeed be labeled in red. We have revised **Figure 1** accordingly to reflect this.
>
> **Bankruptcy and rounds effect definitions**
>
> Thank you for pointing out the need for clearer definitions. We confirm that:
>
> - **Bankruptcy**: Running out of capital (balance falling below the minimum bet amount), which forces immediate game termination—as opposed to voluntary stopping
> - **Rounds Effect**: Quantifies the directional impact of a specific prompt component on game length, calculated as the mean difference between conditions with and without that component
>
> We have updated the manuscript to include these clarifications.
>
> **6400 games and embedding extraction**
>
> Yes, 6,400 games mean LLaMA played 100 times in each of 64 conditions. For SAE feature extraction, we extracted features from the **last token position** of the model's response at the final decision round of each game. Specifically, we used the hidden state at position `[:, -1:, :]` from the model's output, which corresponds to the last generated token before game termination. This position captures the model's final decision state, making it most relevant for analyzing the cognitive patterns leading to different outcomes.
>
> **Safe context and risky context in activation patching**
>
> Safe context and risky context refer to the final decision prompts from games that ended in voluntary stopping and bankruptcy, respectively, from the LLaMA behavioral experiment. The activation patching experiments were validated through 50 independent trials per condition to minimize variance. Error bars representing standard error are included in the figure.
>
> **LLaMA showing similar patterns**
>
> Yes, LLaMA-3.1-8B shows similar patterns as found in Phase 1, although with lower absolute bankruptcy rates. The key finding—that variable betting produces higher bankruptcy than fixed betting—remains consistent across all models including LLaMA (Fixed: 2.62% vs Variable: 6.75%).

---

### Official Review · Reviewer_LQ6B · 2025-11-01

**Soundness:** 2
**Presentation:** 2
**Contribution:** 2
**Rating:** 2
**Confidence:** 4

**Summary:**

This paper investigates whether large language models (LLMs) can exhibit gambling addiction-like behaviors similar to humans. The authors use classical gambling tasks from human experiments to test LLMs and develop quantitative measures to assess addiction-related behaviors. They perform neural circuit analysis using sparse autoencoders to identify underlying mechanisms. Behaviorally, the study reports addiction-like gambling patterns that lead to bankruptcy in LLMs. Through neural network analysis, the authors identify features that distinguish between bankruptcy and self-stopping cases, which they interpret as risky and safe decision-making features.

**Strengths:**

**1.** **Multi-level analysis approach**: The paper combines behavioral measures with neural network-level analysis and intervention, providing insights at both phenomenological and mechanistic levels.

**2.** **Systematic experimental design**: The authors test 64 different experimental conditions, including a comprehensive examination of how information conveyed through prompts may increase or decrease gambling behaviors.

**Weaknesses:**

**1.** **Unclear objective and limited ecological validity**: The paper's objective appears misaligned with its experimental design and data analysis. If the goal is to inform investment applications, the study should use realistic investment scenarios rather than simplified gambling tasks to ensure ecological validity. Conversely, if the aim is to simulate human pathological gambling behaviors, the paper should demonstrate behavioral resemblance to humans at a finer granularity, such as the cognitive algorithms that lead to such sequential gambling behaviors. The current approach falls short of either objective.

**2.** **Problematic definition and conceptualization of irrationality**: The paper conflates gambling behavior with irrationality, which is theoretically unsound. Rationality depends on the reward structure and the agent's objectives. In some experimental conditions, the prompt explicitly sets the goal as "doubling your initial funds to $200", which cannot be achieved without gambling, while avoiding bankruptcy is not explicitly designated as a goal. Under these conditions, risk-seeking behavior—even when it risks bankruptcy—could be rational, analogous to risk-sensitive foraging in starving animals that increase risk-taking to maximize survival probability. To demonstrate irrationality, the authors must show that the agent's behavior violates its stated goals, which they have not established.

**3.** **Questionable practical implications**: The authors suggest their findings are relevant for financial decision-making applications, yet investment inherently involves calculated risk-taking. An LLM that avoids all gambling-like behaviors would not necessarily make a good investor, undermining the practical relevance of the proposed safety concerns.

**4.** **Multiple confounds limiting validity**: Several design and interpretational issues compromise the study's validity. First, the experimental design conflates risk-seeking with irrationality by ensuring that gambling always yields worse expected payoffs than not gambling. This prevents distinguishing between potentially optimal risk-seeking (under appropriate utility functions or reward structures) and genuinely irrational behavior. Second, the "risky" and "safe" features identified in the neural analysis may simply reflect the binary choice of whether to gamble rather than representing generalizable risk-related decision-making mechanisms. Third, the analysis of prompt complexity (Finding 3 in behavioral analysis) confounds complexity with the number of gambling-related prompts, making it impossible to determine which factor drives the observed effects.

**5.** **Insufficient evidence for claimed cognitive biases**: The authors repeatedly claim that LLMs exhibit cognitive illusions such as illusion of control and gambler's fallacy (e.g., Abstract, line 469), but provide no direct empirical evidence for these specific cognitive biases. Similarly, the assertion that "both loss chasing and win chasing exist" appears to merely describe the tendency to increase bets over time. The evidence for human-like cognitive features of gambling addiction in LLMs remains inadequately supported. Apologies if I had missed anything.

**Questions:**

### Q1: Research Objectives and Ecological Validity
**Question:** What is the primary objective of this work—assessing safety risks for financial AI applications, simulating human pathological gambling, or understanding general risk-taking in LLMs? How does the gambling paradigm connect to your stated objective?

**Suggestion:** Redesign experiments to match your objective: either use realistic investment scenarios with varying risk-reward profiles for financial applications, or systematically compare LLM behaviors with established human behavioral patterns at a granular level for cognitive modeling of pathological gambling.

### Q2: Definition and Framework of Irrationality
**Question:** Given that some prompts instruct LLMs to double their funds (unachievable without gambling) while not establishing bankruptcy avoidance as a goal, how do you justify labeling gambling behavior as irrational? Could this represent rational risk-taking analogous to risk-sensitive foraging?

**Suggestion:** (a) Redefine irrationality by demonstrating violations of normative decision-making principles (e.g., violations of expected utility theory, preference reversals, or money pump scenarios) rather than equating risk-seeking with irrationality.  (b) Modify experimental instructions to explicitly balance multiple goals (e.g., "maximize final wealth while avoiding bankruptcy"), and include control conditions where not gambling can achieve the stated goal. (c) Add experimental conditions with positive expected value gambling and varying risk-reward ratios to distinguish rational risk-seeking from genuinely irrational behavior.

### Q3: Confounds in Experimental Design and Interpretation
**Question:** Since gambling always yields worse expected payoffs, how can you separate rational risk-seeking from irrationality? Aren't the "risky" and "safe" neural features simply capturing whether the model chooses to gamble rather than representing generalizable mechanisms? How do you disentangle prompt complexity from the number of gambling-related cues in Finding 3?

**Suggestion:** (a) Include conditions where gambling has positive expected value; (b) Test whether neural features generalize to different decision-making contexts beyond the task currently used; (c) Design a factorial experiment crossing prompt complexity with gambling-related content to independently assess each factor.

### Q4: Evidence for Cognitive Biases
**Question:** What specific experimental manipulations or behavioral signatures demonstrate illusion of control and gambler's fallacy beyond general gambling tendencies? How does observing "both loss chasing and win chasing" constitute evidence for specific psychological mechanisms rather than merely describing increased betting over time?

**Suggestion:** The authors may choose to soften their claims or provide additional supporting results.


### Additional Minor Comments
* A recent related work (Du, 2025, arXiv:2506.22496v1) has also examined LLM gambling-like behaviors and proposed mitigation strategies through training methods. The authors may discuss how their work relates to and differs from this study.

* Figure 2 appears circular: plotting the correlation between the composite irrationality index and bankruptcy when both are derived from the same behavioral sequences. This requires clarification or justification.

* Figure 2 shows 32 points per plot, but the paper describes 64 experimental conditions. Were fixed and variable betting conditions collapsed?

* The redundancy between I_BA and I_EB components in the Irrationality Index definition should be examined. What is their correlation in the actual LLM behavioral data?

* The terms "prompt complexity" and "number of gambling-related prompts" appear to be used interchangeably (e.g., Figure 4), which is confusing and potentially misleading.

* Figure 5: The error bars are not defined in the legend, and there is no statistical test reported for whether changes across streaks are significant.

* The neural network analysis section reports only 211 bankruptcies among 6400 cases (3.3% bankruptcy rate), which is lower than the lowest rate (6.3% for GPT-4.1-mini) reported in Table 2. The omission of this discrepancy from Table 2 appears misleading and requires explanation.

---

> ### Author Response · Authors · 2025-12-02
>
> ## Major Concern 1: Unclear objective and limited ecological validity
>
> We sincerely appreciate your detailed feedback. As you pointed out, we acknowledge that the issues in our Introduction and the gaps in our experimental design obscured the clarity of our research objective.
>
> Our study's primary objective was to identify the **key mechanisms driving LLMs to exhibit irrational behavior**. While LLMs generally exhibit learned rational decision-making, they are known to engage in significantly riskier behaviors under specific conditions. We introduced a slot machine experiment to isolate and verify these drivers within a controlled environment. In doing so, we observed that the causes of irrational behavior in LLMs closely parallel the pathways through which humans make irrational choices.
>
> We have made two major improvements to the manuscript.
>
> First, we have **significantly restructured the paper**. We minimized content not directly related to our core objective (such as financial decision-making contexts) and moved auxiliary findings that did not effectively highlight our main contributions (e.g., prompt complexity) to the Appendix. Additionally, to effectively illustrate how irrational behaviors manifest in practice, we moved the case study—originally in the Appendix—to the main text.
>
> Second, we **added an Investment Choice experiment to demonstrate cross-paradigm consistency**. This experiment has a different structure from the slot machine, offering four options with identical expected losses but different risk profiles:
>
> - Option 1: safe exit with capital return
> - Option 2: 50% win probability, 1.8× payout
> - Option 3: 25% probability, 3.6× payout
> - Option 4: 10% probability, 9.0× payout
>
> Notably, Options 2 and 4 have identical expected losses (-10%), allowing us to isolate risk preference from expected value computation. Results show:
>
> - Option 4 selection increased from 15% (BASE) to 41% (GM condition)
> - Bankruptcy rates nearly doubled from 40.0% (BASE) to 74.6% (Goal-setting)
>
> This cross-paradigm evidence demonstrates that our findings emerge from decision-making mechanisms rather than task-specific features.

---

> ### Author Response · Authors · 2025-12-02
>
> ## Major Concern 2: Problematic definition and conceptualization of irrationality
>
> We sincerely thank the reviewer for this important critique. It allowed us to identify a critical error in our manuscript. We address this in two aspects: first, we corrected the erroneously listed prompts; second, we clarified our definition of irrationality and added additional experiments and analyses to verify it.
>
> **1. Correcting Prompt Discrepancy**
>
> We apologize for the confusion. The prompts listed in the Appendix were mistakenly taken from an early pilot phase designed to explore factors of risk-taking behavior. Those prompts included a fixed goal of doubling one's funds. However, the prompts actually used in the main experiments reported in the paper were **"First, set a target amount yourself, and then make decisions according to that goal."** This prompt allowed us to observe what the model expects from its gambling approach. We will immediately revise the Appendix to reflect the actual experimental setup.
>
> Under this actual setup, the reviewer's concern—that risk-seeking behavior could be a rational adaptation to a high external goal—no longer applies. Models had the freedom to set achievable goals but frequently set unrealistic targets and employed aggressive betting strategies to pursue self-imposed goals.
>
> **2. Strengthening the Definition and Analysis of Irrationality**
>
> **2.1 Theoretical Framework: Three Core Dimensions of Pathological Gambling**
>
> We acknowledge the reviewer's point that our previous conceptualization conflated rational risk-taking with irrationality. To address this, we restructured our analysis based on clinical gambling disorder literature. We adopted the integrative model proposed by Petry (2005) and Grant & Potenza (2006), which identifies three core dimensions of gambling addiction:
>
> 1. **Betting Aggressiveness**: The propensity to make large or escalating bets relative to available resources
> 2. **Self-Regulation Failure**: The inability to adhere to self-imposed limits and rules
> 3. **Cognitive Distortions**: Unrealistic expectations including unattainable goal-setting or probability misjudgment
>
> **2.2 Reconceptualization**
>
> We removed the composite index (Eq.1) and instead analyze individual metrics (I_BA, I_LC, I_EB) separately. Each metric is independently interpretable:
>
> - **I_BA (Betting Aggressiveness)**: Increased 2.3× (0.14→0.31) under variable conditions
> - **I_LC (Loss Chasing)**: Increased 2.7× (0.16→0.42) under variable conditions
> - **I_EB (Extreme Betting)**: Increased 6.4× (0.04→0.23) under variable conditions
>
> Notably, for I_LC, we changed from a binary measurement (whether betting increased after a loss) to a continuous measurement (relative increase in bet-to-balance ratio). This allows for more precise measurement of loss chasing intensity.
>
> **2.3 Measuring Self-Regulation Failure: Goal Escalation**
>
> We observed a "moving target" phenomenon where models receiving goal-setting prompts continuously raised their targets after achieving them:
>
> - BASE/M conditions: goal escalation rate 21-22%
> - G/GM conditions: goal escalation rate 56-59%
>
> This demonstrates that models continue gambling even after achieving stated goals, exhibiting self-regulation failure.
>
> **2.4 Equal Expected Value Experiment**
>
> To further distinguish rational risk preference from genuine irrationality, the Investment Choice experiment offered options with identical expected losses but different risk profiles. Options 2 (50% win rate) and 4 (10% win rate) have identical expected losses (-10%). This design allows us to separate expected value computation from pure risk preference. Results:
>
> - BASE condition: Option 2 selection 61%, Option 4 selection 15%
> - GM condition: Option 2 selection 31%, Option 4 selection 41%
>
> The systematic shift toward the extreme-risk option (Option 4) under goal-setting conditions suggests condition-induced irrationality rather than simple risk preference.

---

> ### Author Response · Authors · 2025-12-02
>
> ## Major Concern 3: Questionable practical implications
>
> Thank you for this insightful point. We understand your concern regarding whether our findings—discovered in a relatively simple setting and single experimental environment—can be generalized. We address this concern from two perspectives.
>
> First, **our analysis of neural representations demonstrates that our findings depend on generalizable experimental contexts rather than merely specific prompts.** Among 8,644 candidate features, 112 (23 safe + 89 risky) exhibited causal effects, and these are associated with abstract decision-making concepts rather than specific prompt tokens. Word-feature analysis showed safe features associate with termination vocabulary (over, ended, done, final, thanks), while risky features associate with goal-pursuit language (goal, target, make, set).
>
> Second, **we demonstrated generalizability through cross-paradigm consistency.** We observed consistent patterns across two different paradigms—the slot machine experiment and the investment choice experiment:
>
> - Both experiments showed variable betting produces higher bankruptcy/losses
> - Both experiments showed goal-setting as the most powerful risk factor
>
> The practical implication of our study holds value precisely because anomalous behaviors in LLMs have already been observed in various fields. We believe our findings possess sufficient empirical value in that we showed similar errors encountered in diverse real-world cases can be reproduced in our simplified setting, and their causes stem from transferable factors such as autonomy and neural representations.
>
> Nevertheless, we acknowledge the validity of your critique regarding 1) the superficial explanation of the financial domain, and 2) the lack of clarity in the definition of irrationality. Therefore, we have revised the Introduction and clearly defined irrationality based on existing human behavioral studies.
>
> ---
>
> ## Major Concern 4: Multiple confounds limiting validity
>
> We apologize for any confusion caused by our previous, somewhat vague use of "gambling-like behavior" as a synonym for negative outcomes.
>
> First, the primary value of our research lies not merely in preventing gambling-like behavior, but in tracking how LLM decision-making shifts under specific conditions. While taking risks is necessary in investment scenarios, identifying the specific conditions that trigger irrational risk-taking versus safe choices is essential for deploying LLMs in decision-making contexts.
>
> Second, our research focuses on self-regulation failure rather than simple risk-taking. Specifically, we observed:
>
> Goal escalation (56-59%): Raising targets and continuing to gamble even after achieving goals.
>
> Extreme betting (6.4× increase): Wagering more than 50% of the balance on a single bet.
>
> Loss chasing (2.7× increase): Increasing bets immediately following losses.
>
> Notably, while LLMs behave rationally under baseline conditions (BASE condition bankruptcy rate: 40.0%), the sharp increase in these patterns under conditions that enhance autonomy (GM condition bankruptcy rate: 77.3%) represents a significant behavioral shift that must be addressed.
>
> Third, through our Equal Expected Value design, we isolated risk preference from expected value maximization. In the Investment Choice experiment, Option 2 and Option 4 have identical expected losses (-10%) but different variances. Therefore, the systematic shift toward Option 4 represents a pure change in risk preference rather than a failure to calculate expected value.

---

> ### Author Response · Authors · 2025-12-02
>
> ## Major Concern 5: Insufficient evidence for claimed cognitive biases
>
> We demonstrated cognitive biases such as illusion of control and gambler's fallacy through 1) quantitative metrics and 2) case studies.
>
> **Direct Evidence Through Case Studies:**
>
> 1. **Illusion of Control**: GPT-4.1-mini stated "betting $5 has given a better chance of winning... smaller bets have higher probability of winning." This represents the belief that one's betting behavior can alter the fixed 30% win probability of a random process.
> 2. **Gambler's Fallacy**: GPT-4o-mini explicitly stated "the slot machine may be due for a win" after three consecutive losses.
> 3. **Loss Chasing**: GPT-4.1-mini increased its bet 9× from $10 to $90, stating "at least recover to my initial fund... the best is to bet the full $90." This directly mirrors the loss-chasing criterion in DSM-5.
> 4. **House Money Effect**: Gemini-2.5-Flash explicitly mentioned "playing with 'house money'" and increased betting from $400 to $900 (+125%).
>
> We understand your point to be that these correlations do not prove a causal relationship regarding whether LLMs actually "experience" cognitive biases. While we agree with this observation, we ask that you consider that our research focuses on analyzing **the triggering conditions under which irrational decision-making increases in settings analogous to human environments**, rather than addressing the question of whether LLMs genuinely undergo these biases. Therefore, we have revised to use more cautious language: "behavioral patterns showing functional similarities to cognitive distortions."
>
> ---
>
> ## Minor
> ### I_BA and I_EB redundancy
> We appreciate the concern regarding redundancy between I_BA and I_EB. To address this concern, in the revised manuscript we **removed the composite index (Eq.1)** and instead analyze each metric individually.
>
> I_BA measures the average investment ratio (mean bet-to-balance ratio across rounds), while I_EB measures the proportion of cases where 50% or more was invested (frequency of extreme betting). The former reflects general betting patterns, while the latter specifically measures "all-or-nothing" decisions that expose the gambler to immediate bankruptcy.
>
> In the revised paper, we report these two metrics independently without combining them into a single composite index:
>
> - I_BA: 0.14 (fixed) → 0.31 (variable)
> - I_EB: 0.04 (fixed) → 0.23 (variable)
>
> This allows for clearer interpretation of the behavioral aspects captured by each metric.
>
> ### "Prompt complexity" terminology
>
> Thank you for this important clarification. We apologize for the confusion. You are correct. The terms "prompt complexity" and "number of gambling-related prompts" were used interchangeably and imprecisely. The actual variable measured was simply "the number of prompts." We have revised the manuscript to use this precise term consistently. Additionally, we have moved this finding to the Appendix as it does not directly support our main objective.
>
> ### Neural analysis bankruptcy rate discrepancy
>
> We apologize for the confusion caused by the lack of specification regarding the model used in this section. To clarify, the neural network analysis was conducted using the **LLaMA-3.1-8B model**, which exhibited a relatively lower bankruptcy rate (3.3%, 211/6400) compared to the closed proprietary models. We have updated **Table 4** to include the experimental results for LLaMA (Fixed: 2.62%, Variable: 6.75%), thereby explicitly resolving this discrepancy.
>
> ### Figure 5 error bars
>
> The error bars in Figure 5 represent standard error across experimental conditions. We have updated the figure legend to include this definition.
>
> ### Related work (Du, 2025)
>
> Thank you for bringing this recent work to our attention. We have added a discussion of Du (2025, arXiv:2506.22496v1) in our revised manuscript. While Du's work also examines LLM gambling-like behaviors and proposes mitigation strategies through training methods, our study differs in three key aspects: (1) we provide a comprehensive behavioral framework grounded in clinical gambling disorder literature, (2) we identify specific triggering conditions (variable betting, goal-setting prompts), and (3) we demonstrate causal neural mechanisms through SAE analysis rather than focusing on training-based interventions.

---

### Author Response · Authors · 2025-12-02
**Summary of Revisions**

We sincerely appreciate your detailed and constructive feedback. As you pointed out, we acknowledge that the issues in our Introduction and the gaps in our experimental design obscured the clarity of our research objective. Thank you for providing the guidance necessary to improve the quality of our manuscript.

Our study's primary objective was to identify the **key mechanisms driving LLMs to set risk-taking goals and execute irrational behaviors**. While LLMs generally exhibit learned rational decision-making, they are known to engage in significantly riskier behaviors under specific conditions. We introduced a slot machine experiment to isolate and verify these drivers within a controlled environment. In doing so, we observed that the causes of irrational behavior in LLMs closely parallel the pathways through which humans make irrational choices.

Specifically, our key findings can be summarized as follows:

1. **LLMs tend to make riskier choices as their degrees of freedom in decision-making increase.** In the slot machine experiment, fixed betting produced bankruptcy rates of 0-13%, while variable betting elevated bankruptcy to 6-48%. All irrationality metrics increased under variable conditions: betting aggressiveness (0.14→0.31, 2.3×), loss chasing intensity (0.16→0.42, 2.7×), and extreme betting (0.04→0.23, 6.4×).
2. **Goal-setting prompts are the most powerful risk factor.** In the investment choice experiment, baseline bankruptcy rate was 40.0%, while Goal-setting (G) elevated this to 74.6%, and combined GM condition reached 77.3%—nearly doubling the rate. Goal escalation rates also increased from 21-22% (BASE/M) to 56-59% (G/GM).
3. **There exist consistent internal features that influence decision-making toward risky or safe choices regardless of the prompt used.** Among 8,644 candidate features, 112 (23 safe + 89 risky) exhibited statistically significant causal effects. Safe features increased stopping rate by +17.8% and decreased bankruptcy rate by -5.7%, while risky features decreased stopping rate by -9.3% to -18.8% and increased bankruptcy rate by +25.1%. Notably, these features exhibit layer-wise specialization: safe features concentrate in early-to-middle layers (L4-L19), while risky features concentrate in later layers (L24-L28).

Motivated by your critique that our objective was unclear, definitions were imprecise, and findings lacked experimental robustness, we have made three major improvements to the manuscript:

**First, we have significantly restructured the paper.** We minimized content not directly related to our core objective (such as financial decision-making contexts) and moved auxiliary findings that did not effectively highlight our main contributions (e.g., prompt complexity) to the Appendix. Additionally, to effectively illustrate how irrational behaviors manifest in practice, we moved the case study—originally in the Appendix—to the main text.

**Second, we provided more refined definitions of irrationality behavior.** We removed the composite index (Eq.1) and instead analyze individual metrics (I_BA, I_LC, I_EB) separately. We also restructured the theoretical framework around three core dimensions from clinical gambling disorder literature (Petry 2005, Grant & Potenza 2006): betting aggressiveness, self-regulation failure, and cognitive distortions. Notably, for I_LC, we changed from a binary measurement (whether betting increased after a loss) to a continuous measurement (relative increase in bet-to-balance ratio) to more precisely capture loss chasing intensity.

**Third, we conducted additional experiments to effectively demonstrate the increase in risk-taking behavior:**

1. **Investment Choice Experiment**: Offered multiple options with identical expected losses but different risk profiles (Option 2: 50% win rate vs Option 4: 10% win rate) to isolate risk preference measurement
2. **Bet constraint analysis**: Tested fixed-variable relationship across betting amount constraints ($10/$30/$50/$70), finding variable betting consistently produced higher bankruptcy across all constraint levels (average +3.3%)
3. **Extended neural analysis**: Expanded analysis from 7 layers (25-31) to all 31 layers (L1-L31), conducted exhaustive search of causal features, and added word-feature correlation analysis

Nevertheless, we would like to emphasize that the core contribution of our research remains unchanged, and this work still represents a differentiated approach within the AI Safety research domain. Through an extensive experimental design, we identified consistent behavioral patterns in LLM responses, and further analyzed their underlying causes from multiple perspectives.

Detailed modifications can be found in our point-by-point responses below. We have also uploaded the revised manuscript, with changes highlighted in **blue** for your convenience. Once again, thank you for deeply reviewing our paper and providing such valuable feedback.

---

### Meta-Review · Area_Chair_97Vj · 2025-12-29

**Summary:**

Drawing from human gambling addiction research, the paper examines whether LLMs show irrational behavior such as failure of self-regulation and cognitive distortions. It concludes that AI systems do present behaviour resembling human gambling addictions.

The reviewers raised significant concerns with the premise, namely that the definition of "irrationality" used in the paper is flawed because some of the behaviours the paper describes as irrational are actually rational given the prompt the AI is passed. The authors acknowledge that this is a "critical error in our manuscript." While the authors did significantly rewrite their paper to address this issue, I believe the rewrites make this a sufficiently different paper that it needs to be re-reviewed.

The second flaw in this paper is the limited applicability of the experiments. All three reviewers point out that the single gambling task being evaluated is insufficient to draw the broad conclusions that the authors would like to draw. A resubmission with more thorough experiments to support (or refute) the claims the authors would like to make is thus warranted.

**Reviewer Concerns:**

Significant changes are required to make the arguments in the paper sound and well-supported by the presented experiments. A re-review is therefore necessary.

**Reviewer Scores:**

I don't think the reviewers would have changed their scores.

---

### Decision · Program_Chairs · 2026-01-26

Reject